# Coherently Driven and Superdirective Antennas

**Alex Krasnok** 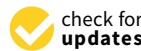

Advanced Science Research Center, City University of New York, New York, NY 10031, USA;
akrasnok@gc.cuny.edu; Tel.: +1(737)781-1203

**Abstract:** Antennas are crucial elements for wireless technologies, communications and power transfer across the entire spectrum of electromagnetic waves, including radio, microwaves, THz and optics. In this paper, we review our recent achievements in two promising areas: coherently enhanced wireless power transfer (WPT) and superdirective dielectric antennas. We show that the concept of coherently enhanced WPT allows improvement of the antenna receiving efficiency by coherent excitation of the outcoupling waveguide with a backward propagating guided mode with a specific amplitude and phase. Antennas with the superdirectivity effect can increase the WPT system's performance in another way, through tailoring of radiation diagram via engineering antenna multipoles excitation and interference of their radiation. We demonstrate a way to achieve the superdirectivity effect via higher-order multipoles excitation in a subwavelength high-index spherical dielectric resonator supporting electric and magnetic Mie multipoles. Thus, both types of antenna discussed here possess a coherent nature and can be used in modern intelligent antenna systems.

**Keywords:** antennas; coherently driven antennas; superdirective antennas

---

## 1. Introduction

An antenna is a key element for many vital wireless technologies, including communications and power transfer [1]. The first antennas emerged with the discovery of electromagnetic waves by Hertz in 1888, and since then have developed alongside human civilization, often being a catalyst for its development. A plethora of antennas have been invented in the radio and microwave frequency ranges, including microstrip antennas [2–4], reflector antennas [1,5], and dielectric antennas [6,7] to mention just a few. More recently, so-called nanoantennas or antennas operating in the optical frequency range have been invented, which have become irreplaceable elements for quantum optics and communications on a chip [8–14].

While wireless communications are rather established, the wireless power transfer (WPT), proposed at the beginning of the 20th century by Tesla is experiencing a rebirth. It was caused by an experimental demonstration by Kurs et.al. [15] that the WPT efficiency can be drastically enhanced when the power is transferred via resonant coupling. In that paper, wireless energy transfer between two metallic coils over the distance of 2 m with 45% efficiency in the kHz range via strongly coupled magnetic resonances was demonstrated [15]. These fascinating results have given rise to many novel technologies, including implanted devices, electric vehicles, and consumer electronics [16]. Since then, significant research efforts have been devoted to exploring the ways to achieve as high WPT efficiency as possible [16], whereas the majority of research has been concentrated on optimization of electromagnetic resonators' geometry, surrounding material, their relative arrangement.

On another hand, the design of highly directive antennas is a significant problem in the theory of antennas, which has been concerning researchers and engineers for a long time. The importance



of highly directional antennas for wireless interconnections and power transfer can be understood considering the Friis equation (for matched antennas)

$$P_{\text{rec}} = \eta_r \eta_t D_r D_t |\mathbf{a}_r \cdot \mathbf{a}_t^*| \frac{\lambda_0^2}{(4\pi d)^2} P_{\text{tr}} \tag{1}$$

which says that the transmitted power ($P_{\text{rec}}$) scales as product of receiver and transmitter antennas' directivities ($D_r D_t$) and also depends on their radiation efficiencies ($\eta_r$ and $\eta_t$), operation wavelength ($\lambda_0$), separation distance ($d$), relative polarization ($\mathbf{a}_r \cdot \mathbf{a}_t^*$), and total transmitted power ($P_{\text{tr}}$). Hence, if directivities of the both antennas are the same ($D$) then the transmitted power scales as $D^2$, and hence can be significantly enlarged by using more directive antennas. There are various approaches to achieve a sufficiently high directivity for practical purposes relying on large geometrical dimensions, including Yagi-Uda antenna, lens antenna, leaky-wave antenna, and others. All these approaches rely on the radiated (received) wave propagation in the antenna structure and, as a result, these antennas have a large size ($l >> \lambda_0$) in at least one direction. However, many applications require an antenna to be directive and compact ($l < \lambda_0$) at the same time. Such antennas whose size is smaller than the operation wavelength in all three directions and directivity much larger than the directivity of a short dipole antenna (1.5) are called superdirective antennas [17–24]. Superdirective antenna operation relies on creating rapidly spatially oscillating currents in a subwavelength area, which leads to excitation of higher multipoles [17,24]. As a result, the antenna becomes directive despite its subwavelength volume. Usually the superdirectivity regime is achieved in arrays of short dipoles, which are fed with a specific amplitude and phase. This approach appeared to be unstable and energy-consuming and, as a result, did not find a widespread practical application.

In this paper, we present our recent achievements in coherently enhanced wireless power transfer and superdirective antennas [24–26]. Figure 1a demonstrates schematically the operation principle of coherently enhanced wireless power transfer. Any WPT system consists of at least two antennas: transmitting and receiving ones. The transmitting antenna (right) radiates radio waves toward receiving antenna (left), which receives some power. We have demonstrated that the same receiving antenna driven by an additional wave (red wave) can either receive more power that it did without the coherent excitation or become more stable to changes in environment or relative arrangement. The next section in dedicated to such coherently enhanced WPT antennas. Figure 1b demonstrates the superdirective dielectric antenna consisting of a subwavelength spherical dielectric resonator (shape can be different) with a small notch on its surface and fed by a short dipole source placed in the notch. Inhomogeneous near field of the dipole source excites high-order multipoles in the resonator which radiate coherently and can form the superdirectivity regime. This approach is discussed in the Section 3.

We note that both types of antennas discussed here essentially rely on coherent effects. The coherently enhanced antenna allows improvement of the antenna receiving efficiency by coherent excitation of the outcoupling waveguide with a backward propagating guided mode with a specific amplitude and phase, whereas the superdirective antenna possesses an increased directivity due to coherent radiation of higher-order multipoles. The both antennas are experimentally realized [24,25].

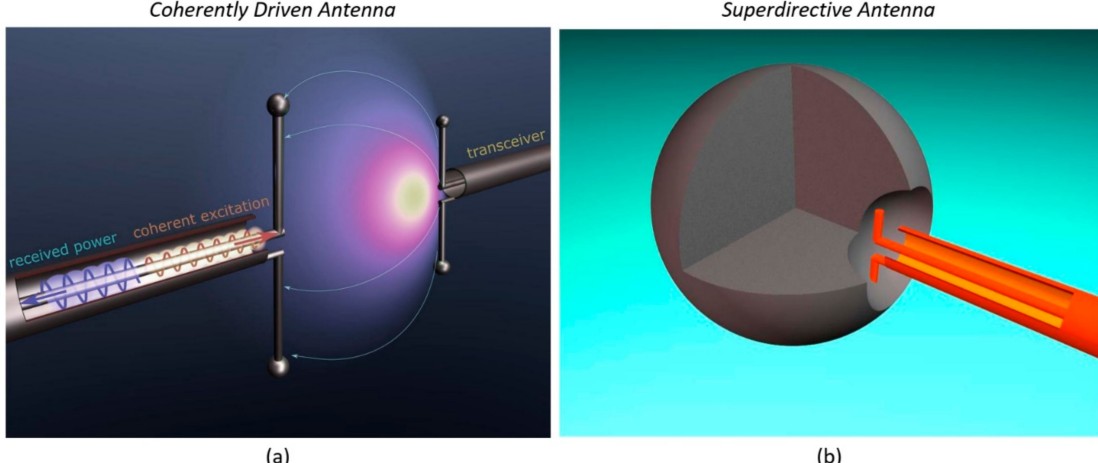

**Figure 1.** (**a**) Operation principle of coherently enhanced wireless power transfer: a transmitting antenna (right) radiates radio waves to the receiving antenna (left) driven by an additional wave sent from the circuit to the antenna (red wave). (**b**) Superdirective dielectric antenna: subwavelength spherical dielectric resonator (shape can be different) with a small notch on its surface fed by a short dipole source placed into the notch. Inhomogeneous near field of the dipole source excites high-order multipoles in the resonator which radiate coherently and can form the superdirectivity regime. Reprinted with permission from [24,25].

## 2. Coherently Enhanced Wireless Power Transfer

The concept of coherently enhanced WPT is based on the fundamental property of the wave nature of electromagnetic field-interference. Only recently it has been clearly emphasized that electromagnetic processes such as absorption and scattering may be effectively controlled via coherent spatial and temporal shaping of the incident electromagnetic field. For example, a coherent perfect absorber (CPA) is a linear electromagnetic system in which perfect absorption of radiation is achieved with two or more incident waves, creating constructive interference inside an absorbing structure. In this section, we show that the same principle that underlies the operation of CPAs can be employed to improve the efficiency of WPT systems [25]. More specific, by coherent excitation of the receiving antenna (see Figure 1a) with an auxiliary signal, tuned in sync with the impinging signal from the transmitting antenna, it is possible to enhance the efficiency and robustness of the WPT system. This additional signal improves energy transfer through constructive interference with the impinging wave, compensating any imbalance in the antenna coupling without having to modify the load.

Since the concept of coherently enhanced WPT is relative to CPA effect, first we discuss the main aspects of coherent perfect absorption. Coherent perfect absorption generalizes the concept of ordinary perfect absorbers to systems with two or more excitation channels [27–31]. This phenomenon can be illustrated by consideration of a two-port planar structure, Figure 2. Excitation of this resonator from one channel by a wave $s_1^+$ results in partial transmission ($s_1^+ t_{12}$) and reflection ($s_1^+ r_{11}$). However, excitation by two waves from both channels ($s_1^+$ and $s_2^+$) result in outgoing waves in both channels ($s_1^-$ and $s_2^-$) and can be described by the $\hat{S}$ matrix, which links waves in inputs and outputs, $\begin{pmatrix} s_1^- \\ s_2^- \end{pmatrix} = \hat{S} \begin{pmatrix} s_1^+ \\ s_2^+ \end{pmatrix}$,

where $\hat{S} = \begin{pmatrix} r_{11} & t_{12} \\ t_{21} & r_{22} \end{pmatrix}$. Here $s_i^+$ and $s_i^-$ respectively denote the input and output wave amplitudes in the i-th channel; $r_{ii}$ are the reflection coefficients in each port, and $t_{ij}$ are the transmission coefficients. This $\hat{S}$ matrix has the following eigenvalues $d_{1,2} = \frac{1}{2}\left(r_{11} + r_{22} \pm \sqrt{r_{11}^2 - 2r_{11}r_{22} + r_{22}^2 + 4t_{12}t_{21}}\right)$ and eigenvectors $s_{1,2} = \left\{1, r_{11} - r_{22} \mp \sqrt{r_{11}^2 - 2r_{11}r_{22} + r_{22}^2 + 4t_{12}t_{21}}/2t_{21}\right\}$. If $\hat{S}$-matrix is constrained by optical reciprocity [32], then $t_{12} = t_{21} = t$. If in addition the two-port structure is symmetric under

mirror reflection, $r_{11} = r_{22} = r$, the corresponding eigenvalues and eigenvectors become $d_{1,2} = r \pm t$ and $s_{1,2} = \{1, \pm1\}$ representing symmetric and antisymmetric inputs of equal intensity.

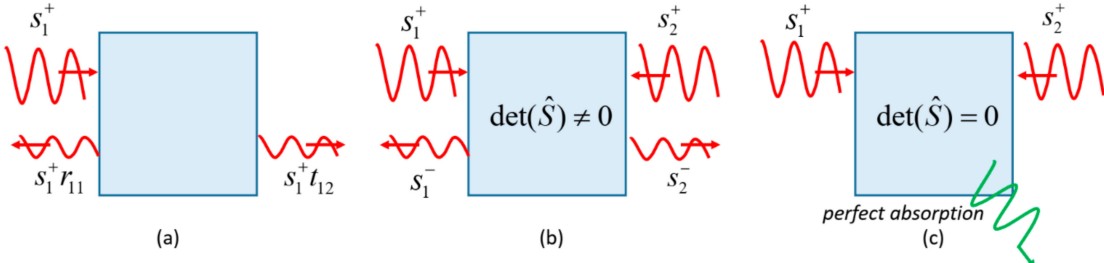

**Figure 2.** System with two channels, for example, dielectric resonator slab. (**a**) Excitation of this system from one channel by a wave $s_1^+$ results in partial transmission ($s_1^+ t_{12}$) and reflection ($s_1^+ r_{11}$). (**b**) Excitation by two waves from both channels ($s_1^+$ and $s_2^+$) result in outgoing waves in both channels ($s_1^-$ and $s_2^-$) if $\det(\hat{S}) \neq 0$. (**c**) Tailoring the structure geometry (thickness) and adding loss causes CPA regime ($\det(\hat{S}) = 0$), when all incoming energy gets dissipated. Changing of relative amplitude or phase of $s_1^+$ and $s_2^+$ causes changing of mutual absorption and transmission.

Next, if the system is lossless, all zeros of $d_{1,2} = r \pm t$ (i.e., $\det(\hat{S}) = 0$) are located in the upper complex frequency plane and we always have some outgoing waves in both channels ($s_1^-$ and $s_2^-$). However, the situation drastically changes when we add a certain amount of material losses. In this case, it is possible to get the S-matrix zero ($\det(\hat{S}) = 0$) at the real axis, when $r \pm t = 0$. Then excitation of the resonator by the corresponding CPA eigenmode $s_{1,2} = \{1, \pm1\}$ will lead to complete absorption, i.e., CPA effect, Figure 2c. It turns out that the absorption in CPA is highly sensitive to variations of the excitation conditions due to its coherent nature. Changing of relative amplitude or phase of $s_1^+$ and $s_2^+$ causes changing of absorption, which can be characterized by joint absorption [33] $\mathcal{A} \equiv 1 - \frac{|s_1^-|_2 + |s_2^-|^2}{|s_1^+|_2 + |s_2^+|^2}$. Here, $|s_1^+|_2 + |s_2^+|^2$ is proportional to the total input intensity, and $|s_1^-|_2 + |s_2^-|^2$ to the total output intensity; $\mathcal{A} = 1$ in the CPA regime [27,29,33].

In our recent paper we have shown that the same principle that underlies the operation of a CPA can be employed for improving the efficiency of WPT [25]. More specifically, we have shown that there is a possibility to improve the receiving efficiency of an antenna by coherent excitation of the outcoupling waveguide with a backward propagating guided mode with specific properties, Figure 1a. Figure 3a,b demonstrate this idea. Figure 3a shows the traditional WPT system consisting of transmitter antenna (right) and receiver antenna (left). Next, we assume that the receiver antenna is driven coherently by an auxiliary wave (shown in blue), Figure 3b. We have shown that the analysis of this system can be performed in the framework of the temporal coupled mode theory (TCMT) [34–36]. This analysis assumes that the receiving antenna has a single mode with the real eigenfrequency $\omega_0$ and the mode amplitude $a$, normalized such that $|a|^2$ is the energy of the mode. The dipole antenna couples to the waveguide and free space radiation with the coupling constants $|\kappa\rangle = \{\kappa_w, \kappa_f\}$. The excited antenna mode can decay to both channels with the total dumping rate, $1/\tau = 1/\tau_w + 1/\tau_f$. Next, the antenna mode is excited by is the vector of input amplitudes $|s_+\rangle$ consisting of the field of the transceiver $s_f$ and the auxiliary field of coherent excitation $s_w^+$. The results this analysis gives us the following formula for the amplitude of the received field ($s_w^-$, extracted field) [25].

$$s_w^- = -s_w^+ + \kappa_w \frac{\kappa_w s_w^+ + \kappa_f s_f}{i(\omega - \omega_0) + 1/\tau} \qquad (2)$$

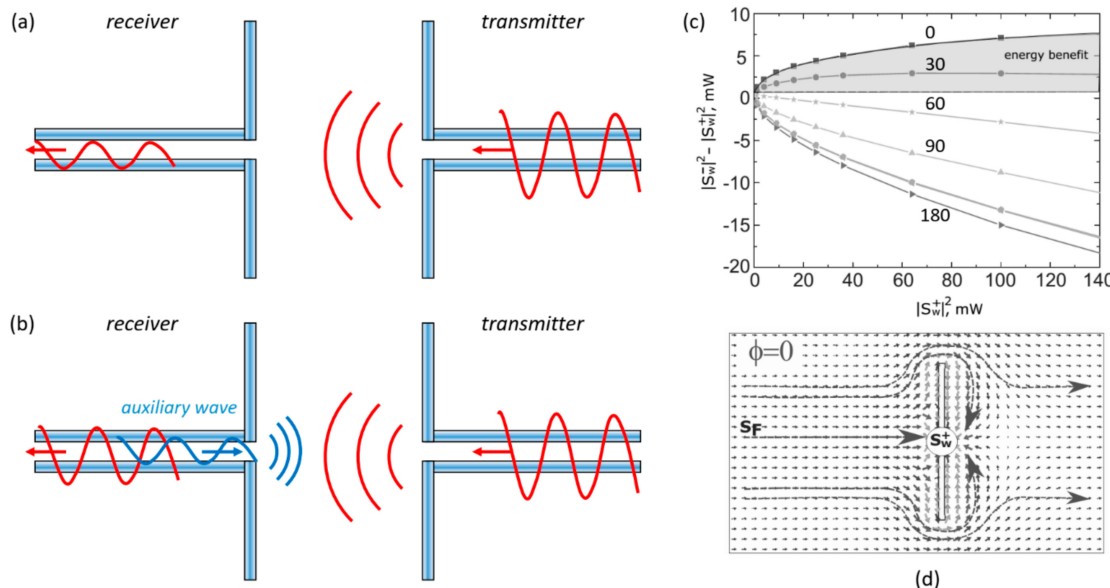

**Figure 3.** (**a**) Traditional wireless power transfer (WPT) system consisting of transmitter antenna (right) and receiver antenna (left). (**b**) Coherently enhanced wireless power transfer system with the receiver antenna driven coherently by an auxiliary wave (shown in blue). (**c**) Net extracted energy as a function of the auxiliary signal intensity for different relative phase values (from 0 to 180 grad). Filled area indicates the region of energy benefit, where the coherently assisted energy balance exceeds that for $s_w^+ = 0$. (**d**) Poynting vector distribution around the antenna with the auxiliary signal $s_w^+$ and relative phase of 0 deg. (**c**,**d**) Reprinted with permission from [25].

The typical results that this equation yields for the net extracted energy are shown in Figure 3c. The area where the net extracted energy exceeds that for $s_w^+ = 0$ (filled area) corresponds to the energy benefit. In these regimes, the antenna revives more energy from the transceiver than it does without coherent excitation even after subtraction of this additional energy. It is interesting that in such a regime of positive net extracted energy the Poynting vector distribution around the antenna demonstrates many flow lines ending by the dipole antenna, see Figure 3d. Otherwise, when the relative phase between $s_w^+$ and $s_f$ is 180 deg., there are a few of Poynting vector lines flowing into the antenna that operates in the radiation regime (radiates more than receives).

In a practical WPT device, the amplitude and phase of the additional coherent signal can be controlled in real time to adjust the antenna as a function of changes in the environment, temperature changes in the load, and distance of the transmitter.

## 3. Superdirective Antennas

Usually, small antennas like dipole electric or magnetic source possess weak directivity close to that of a point dipole (~1.5) [37]. Here, the directivity parameter is defines as $D = 4\pi \max[p(\theta, \varphi)]/P_{tot}$, where $P_{tot}$ is the total radiated power and $p(\theta, \varphi)$ is the power density of radiation in the direction $(\theta, \varphi)$. Typical directive antennas such as Yagi-Uda, lens antennas, refractive antennas rely on wave propagation/refraction and hence have dimensions much larger than the operation wavelength, $l \gg \lambda_0$. However, as was mentioned in the introduction, there are many application areas where electrically small antennas ($l < \lambda_0$) that are very directive are required. This caused intensive research on electrically small radiating systems whose directivity exceeds significantly that of a dipole. Such radiative systems were called superdirective [17–24]. Superdirectivity regime is of a particular interest for space communications and radio astronomy. Moreover, achieving high radiation directivity is also important for actively studied optical nanoantennas [38–40].

Initial attempts to achieve superdirectivity regime have been made in the microwave frequency range where it has been demonstrated that antenna arrays can work in this regime in a very narrow

frequency range for a sophisticated system of phase shifters [17,19,37,41–43]. Physically, operation principles of superdirective antennas rely on creating rapidly spatially oscillating currents in a subwavelength area, which leads to excitation of higher multipoles [17,24]. Coherent interference of field radiated from all phased dipole or, more generally, multipoles leads to the formation of a narrow lobe of power pattern. The antenna becomes directive despite its subwavelength volume. Later, the appearance of the concept of metamaterials [44]—artificially engineered materials with electromagnetic properties at will—caused a new research interest in electrically small directive antennas [45–48].

Recent studies on high-index dielectric antennas made of resonators possessing both resonant electric and magnetic responses made it feasible to invent new efficient antennas in microwave [24,49,50] and optics [13,23,51–54]. It was shown that such subwavelength high-index (usually $n \geq 4$) dielectric spherical resonators possess electric and magnetic Mie resonances [24,49,50,55,56], which can be utilized for antenna designing. Although so-called dielectric resonator antennas have been known for decades [57,58], in these studies the critical role of the magnetic Mie resonance has been uncovered first in optics [10,59–61] and later in microwaves [49] and THz due to the scalability of the Maxwell's equations.

A very well-known system of electric and magnetic dipoles radiating coherently with the same phase is called Huygens source [1,62,63]. As a result, this source consisting of two multipoles exhibits nearly twice the higher directivity than that of a single electric dipole. It has been demonstrated, both in optics and microwaves, that a simple high-index dielectric resonator (spherical or cylindrical) can operate in the Huygens source regime [51,56]. This leads us to the idea that if many multipoles are excited in a system, the directivity of this system can be much greater. At the same time, the size of this dielectric antenna may be sufficiently subwavelength simply because of its high refractive index. This idea has been suggested and realized in our recent works [12,23,24].

Our approach to superdirective emission is based on making a small notch on a subwavelength high-refractive index spherical dielectric resonator supporting electric and magnetic Mie resonances and placing a dipole source inside this notch, Figure 4a,b. This notch breaks the symmetry and increases the contribution of higher-order multipoles [24,26]. The realized antenna consists of a dielectric spherical resonator of size ~$\lambda_0/2.5$ and operates at ~17 GHz. As a material of the resonator was MgO-TiO$_2$ ceramic spheres characterized at microwaves by permittivity of ~16 and small loss tangent. This spherical resonator made of MgO-TiO2 with the radius 5 mm exhibits lowest Mie resonance at the frequency of ~14 GHz. The radius of the semi-spherical notch was 2 mm, the length of the dipole source was 1.5 mm. Note also that theoretically, there are no limitations on the antenna size and it can be made smaller by using dielectrics with the higher refractive index. However, practically, it is limited by its bandwidth and by the size of the feed waveguide. The dipole is fed by a coaxial cable, see Figure 4a. Figure 4c shows the results of numerical simulations of the antenna's directivity and radiation efficiency. We observe a large directivity (~11) for such small antenna. This analysis and the analysis of the effective antenna aperture clearly demonstrates that the antenna operates in the superdirectivity regime. The power pattern of this antenna has been measured in an anechoic chamber and the results are presented in Figure 4d,e on the left panel. We see that these results are in a good agreement with the numerical simulations. Remarkably, this antenna exhibits so-called beam steering effect preserving the superdirectivity regime when the dipole source experiences a small shift to the left/right, see Figure 4e.

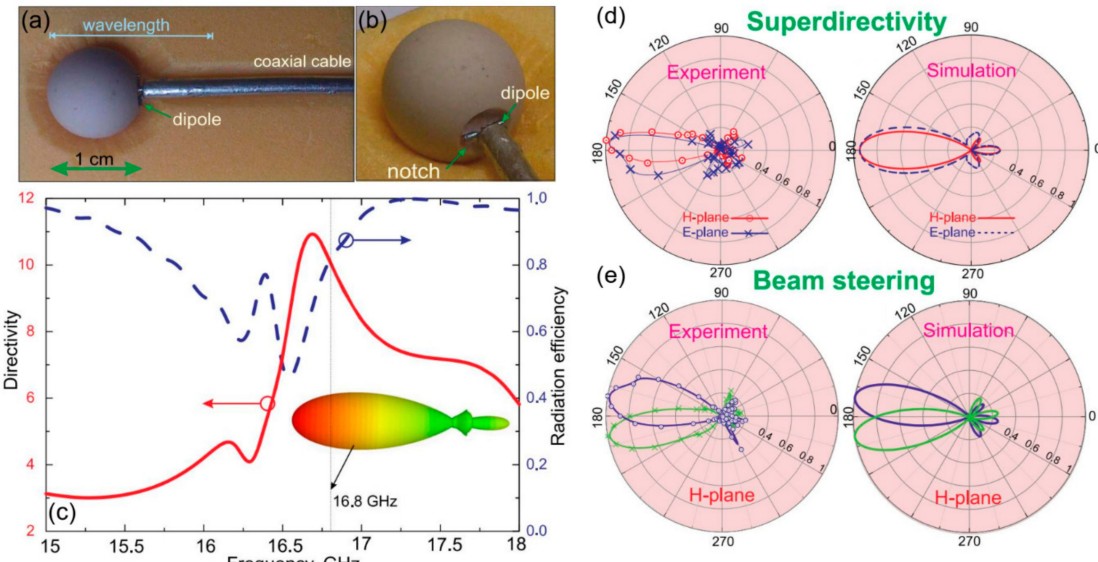

**Figure 4.** Superdirective dielectric antenna. (**a**,**b**) Photos of the realized antenna. The antenna consists of a dielectric spherical resonator of size $\sim\lambda_0/2.5$ with a small notch and a short dipole source placed inside the notch (**b**). The dipole fed by a coaxial cable. (**c**) Results of numerical simulations of the antenna's directivity and radiation efficiency. (**d**,**e**) Results of experimental realization in an anechoic chamber of the superdirectivity effect (**d**) and the beam steering effect (**e**) and their comparison with numerical simulation results. Reprinted with permission from [24].

Coherent nature of the superdirectivity effect in this dielectric antenna can be revealed by multipole decomposition technique, Figure 5, which consists of the following. First, we use numerical simulation to calculate the internal electric and magnetic field distributions. Next, knowing these fields one can calculate the distributions of densities of (bound) carriers $\rho = 1/(4\pi)\mathrm{div}\mathbf{E}$ and currents $\mathbf{j} = c/(4\pi)(\mathrm{rot}\mathbf{H} + ik\mathbf{E})$. Finally, using these sources one can calculate the spherical harmonic electric and magnetic coefficients $a_E(l,m)$ and $a_M(l,m)$, which characterize the electrical and magnetic multipole moments [64]:

$$a_E(l,m) = \frac{4\pi k^2}{i\sqrt{l(l+1)}} \int Y_{lm}^* \left( \rho \frac{\partial}{\partial r}[r j_l(kr)] + \frac{ik}{c}(\mathbf{r}\cdot\mathbf{j}) j_l(kr) \right) d^3r \tag{3}$$

$$a_M(l,m) = \frac{4\pi k^2}{i\sqrt{l(l+1)}} \int Y_{lm}^* \mathrm{div}\left( \frac{\mathbf{r}\times\mathbf{j}}{c} \right) j_l(kr) d^3r \tag{4}$$

where $Y_{lm}$ are spherical harmonics of order $(l > 0, 0 \geq |m| \leq l)$, $c$ is the light velocity, and $k = 2\pi/\lambda_0$.

Figure 5a,b demonstrate the results of the multipole decomposition of the antenna in the superdirective regime. Here, the magnitude (upper row) and phase (lower row) of the most contributing electric (a) and magnetic (b) multipole moments are presented. Multipole coefficients providing the largest contribution to the antenna directivity are highlighted by red circles. We see that despite the small antenna size, it supports multipoles of high order, including electric and magnetic dipole ($l = 1$), magnetic quadrupole ($l = 2$), octupole ($l = 3$) and so on.

We have also demonstrated that this superdirectivity regime is accompanied by a significant increase of the effective near field zone of the antenna compared to that of a point dipole for which the near zone radius is $\sim\lambda_0/2\pi$ [26].

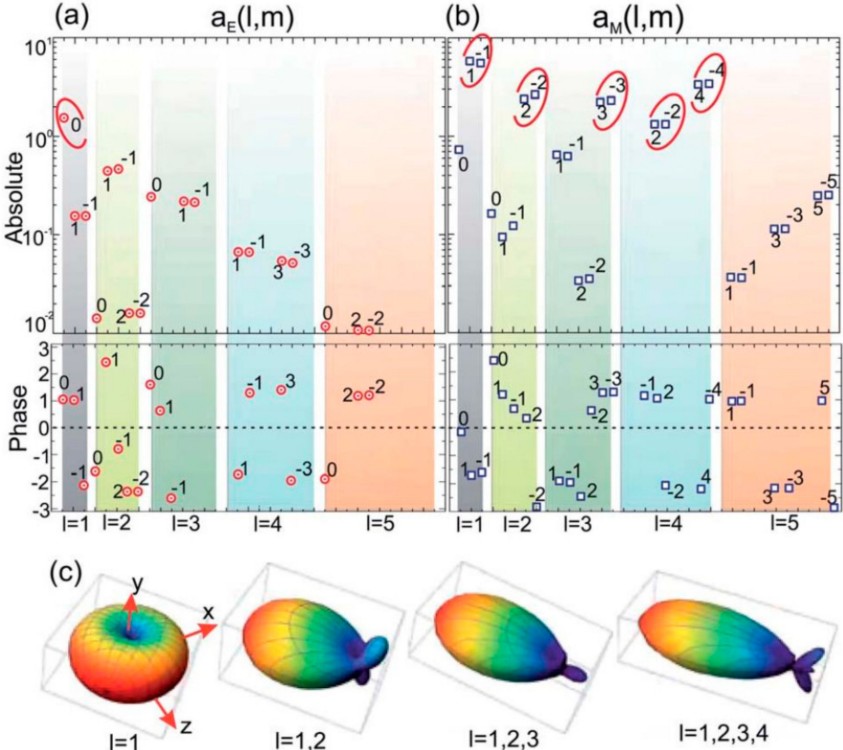

**Figure 5.** Magnitude (upper row) and phase (lower row) of the most contributing (**a**) electric and (**b**) magnetic multipole moments. Multipole coefficients providing the largest contribution to the antenna directivity are highlighted by red circles. (**c**) Dependence of the radiation pattern of dielectric superdirective antenna on the number of considered multipoles (the dipole source is oriented along the z-axis). Reprinted with permission from [23].

The results of multipole decomposition allow us to retrieve the power pattern on the antenna and study the role of each multipole to the superdirectivity regime formation. Figure 5c demonstrates the dependence of the radiation pattern of the dielectric superdirective antenna on the number of considered multipoles (the dipole source is oriented along the z-axis). This result shows that the directivity grows as higher order multipole terms are added to the response. We see that the coherent emission from many high-order multipoles is crucial for the superdirectivity formation.

Finally, we note that despite an impedance matching a superdirective antenna usually being a nontrivial task due to its small size, the reported dielectric antenna demonstrates a good matching with a low level of return losses (0.1 at 16.8 GHz) [24]. In the paper dedicated to experimental realization of this antenna [24] this matching at the superdirective operation regime has been attributed to the classical analogy of the Purcell effect [52]. Remarkably, this matching is not related to dissipative losses and additional matching/symmetrizing devices such as a "balun" [1] are needed.

## 4. Conclusions

Both types of antenna discussed here, coherently driven and superdirective all-dielectric can be utilized in modern intelligent antenna systems in not only radio and microwaves but also in optics. This paper has been dedicated to the analysis of our recent works in this area. We have shown that the concept of coherently enhanced WPT allows improvement of the antenna receiving efficiency by coherent excitation of the outcoupling waveguide with a backward propagating guided mode with a specific amplitude and phase. Antennas with the superdirectivity effect can increase the WPT system's performance in another way, through tailoring of radiation diagram via engineering antenna multipoles excitation and interference of their radiation. We have demonstrated a way to achieve the

superdirectivity effect via higher-order multipoles excitation in a subwavelength high-index spherical dielectric resonator supporting electric and magnetic Mie multipoles.

**Funding:** This research received no external funding.

**Conflicts of Interest:** The authors declare no conflict of interest.

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
