# Peer review of "Coherently Driven and Superdirective Antennas"

_electronics, doi:10.3390/electronics8080845_

Round 1
Reviewer 1 Report
Thank you for working on coherently driven and directive antennas. My comments on this work are as follows:
(1) all figure quality needs to be improved considerably.
(2) figure 4 part (d) and (e) is very blurred and need to be replotted.
(3) Make all the traces colored in the plots.
(4) Since in the first paragraph of intro., the author discussed microstrip and wideband antennas. following references need to introduce: (a) Electronics 2019, 8(2), 158 (b) Sensors 2018, 18(10), 3330
(5) Please provide a table of comparison and show your work superiority there.
(6) The author has performed this work in his majority of previous papers as referred, so what I understand that please place a detailed paragraph and show that what is the difference between both works.
(7) Finally, is there some setup arranged and have some measurements?
Author Response
Comment: All figure quality needs to be improved considerably
Response: I have improved the quality of all the figures.
Comment: figure 4 part (d) and (e) is very blurred and need to be replotted
Response: The quality of the figure is increased
Comment: Make all the traces colored in the plots.
Response: This comment is addressed
Comment: Since in the first paragraph of intro., the author discussed microstrip and wideband antennas. Following references need to introduce: (a) Electronics 2019, 8(2), 158 (b) Sensors 2018, 18(10), 3330
Response: Both references have been added to the text
Comment: Please provide a table of comparison and show your work superiority there
Response: Thank you for this comment! Actually, the reason for this review paper was not to compare with other works but rather to summarize new and unusual approaches to the antenna design, namely based on auxiliary coherent excitation and higher-order mode excitation.
Comment: The author has performed this work in his majority of previous papers as referred, so what I understand that please place a detailed paragraph and show that what is the difference between both works.
Response: I have addressed this comment by adding the following text (see p.3):
“We note that both types of antennas discussed here essentially rely on coherent effects. The coherently enhanced antenna allows improvement of the antenna receiving efficiency by coherent excitation of the outcoupling waveguide with a backward propagating guided mode with a specific amplitude and phase, whereas the superdirective antenna possesses an increased directivity due to coherent radiation of higher-order multipoles.”
Comment: Finally, is there some setup arranged and have some measurements?
Response: Thank you for this comment. Sure, both antennas are realized and measured. The results are presented in Refs. [Krasnok, A.; Baranov, D.G.; Generalov, A.; Li, S.; Alù, A. Coherently Enhanced Wireless Power Transfer. Phys. Rev. Lett. 2018, 120, 143901.; Krasnok, A.E.; Filonov, D.S.D.S.; Simovski, C.R.C.R.; Kivshar, Y.S.Y.S.; Belov, P.A.P.A. Experimental demonstration of superdirective dielectric antenna. Appl. Phys. Lett. 2014, 104, 133502].
I have addressed this comment by adding the following text: “The both antennas are experimentally realized in Refs. [18,19].”, see p. 3.
Reviewer 2 Report
The author proposed an interesting concept of coherently enhanced wireless power transfer and superdirective antennas. The precise and accurate analysis of proposed work is provided. The author also claims that improvement in the antenna receiving efficiency can be achieved by coherent excitation of the outcoupling waveguide with a backward propagating guided mode with a specific amplitude and phase. The overall paper is well written and technically sound. However, there are few minor concerns that needs to be addressed before considering the manuscript for the publication.
1. Be consistent with equation’s format and alignments.
2. Provide reference to the Fig 1 if it is taken from somewhere.
3. How much proposed antenna size is reduced and what are the limitations (minimum size) with respect to size.
4. Page 3 row 62, remove the word “areas”
5. Page 4 row 115, remove the gap between the line
6. Figure 2 and 3 are blur.
7. Why any shape of dielectric resonator is applicable? Need to elaborate or provide at least some simulation results to justify the claim.
8. Lack of Consistency in References format.
Author Response
Comment: The author proposed an interesting concept of coherently enhanced wireless power transfer and superdirective antennas. The precise and accurate analysis of the proposed work is provided. The author also claims that improvement in the antenna receiving efficiency can be achieved by coherent excitation of the outcoupling waveguide with a backward propagating guided mode with a specific amplitude and phase. The overall paper is well written and technically sound.
Response: I would like to thank the referee for high evaluation of my work!
Comment: Be consistent with the equation’s format and alignments.
Response: I have addressed this comment by trying to make all equations in the same format.
Comment: Provide reference to the Fig 1 if it is taken from somewhere.
Response: I have addressed this comment.
Comment: How much proposed antenna size is reduced and what are the limitations (minimum size) with respect to size.
Response: The size of the reported antenna is ~lambda/2. The size can be reduced even more by utilizing dielectrics with the higher refractive index. Theoretically, there are no limitations on the antenna size. However, practically it is limited by its bandwidth and by the size of the feed waveguide. I have added the following sentence to the text to address this comment:
“Note also that theoretically, there are no limitations on the antenna size and it can be made smaller by using dielectrics with the higher refractive index. However, practically it is limited by its bandwidth and by the size of the feed waveguide.” (p.8)
Comment: Page 3 row 62, remove the word “areas”
Response: Done
Comment: Page 4 row 115, remove the gap between the line
Response: Done
Comment: Figure 2 and 3 are blur
Response: I have changed the figures accordingly
Comment: Why any shape of dielectric resonator is applicable? Need to elaborate or provide at least some simulation results to justify the claim.
Response: Thank you for this comment! I have removed that statement from the text (see p.8).
Comment: Lack of Consistency in References format
Response: I have corrected the References to get them in the same format.
Reviewer 3 Report
This is a very nice review of coherently driven and superdirective antennas. I don't see any major issues in this manuscript. I recommend to accept it after addressing some minor issues.
The superdirective antenna is an electrically small antenna, which is usually not easy for impedance matching. Could you add some information about the S11 of the superdirective antenna, as well as the method of impedance matching?
The legend in figure 3(c) is very small, can you improve and make it easier to read?
Author Response
Comment: This is a very nice review of coherently driven and superdirective antennas. I don't see any major issues in this manuscript. I recommend to accept it after addressing some minor issues.
Response: I thank the referee for such a high evaluation of my work!
Comment: The superdirective antenna is an electrically small antenna, which is usually not easy for impedance matching. Could you add some information about the S11 of the superdirective antenna, as well as the method of impedance matching?
Response: The referee is absolutely right saying that this is an important issue. To address this comment we have added the following text: (see p.10)
“Finally, we note that despite that usually an impedance matching of a superdirective antenna is a nontrivial task due to its small size, the reported dielectric antenna demonstrates a good matching with a low level of return losses (0.1 at 16.8 GHz)[15]. In Ref. [15] this matching at the superdirective operation regime has been attributed to the classical analogy of the Purcell effect[49]. Remarkably, this matching is not related to dissipative losses and additional matching/symmetrizing devices such as a “balun”[1] are needed.”
Comment: The legend in figure 3(c) is very small, can you improve and make it easier to read?
Response: We have changed the figure accordingly.
Round 2
Reviewer 1 Report
All my concerns are met. the author carefully responded to my comments and introduced in the manuscript properly.